# Manganese-enhanced magnetic resonance imaging reveals light-induced brain asymmetry in embryo

**Elena Lorenzi[1]\*, Stefano Tambalo[1], Giorgio Vallortigara[1], Angelo Bifone[2,3]**

[1]Center for Mind/Brain Sciences, University of Trento, Piazza Manifattura, Rovereto, Italy; [2]Center for Neuroscience and Cognitive Systems @ UniTn, Istituto Italiano di Tecnologia, Rovereto, Italy; [3]Department of Molecular Biotechnology and Health Sciences, University of Torino, Torino, Italy

**Abstract** The idea that sensory stimulation to the embryo (in utero or in ovo) may be crucial for brain development is widespread. Unfortunately, up to now evidence was only indirect because mapping of embryonic brain activity in vivo is challenging. Here, we applied for the first time manganese enhanced magnetic resonance imaging (MEMRI), a functional imaging method, to the eggs of domestic chicks. We revealed light-induced brain asymmetry by comparing embryonic brain activity in vivo of eggs that were stimulated by light or maintained in the darkness. Our protocol paves the way to investigation of the effects of a variety of sensory stimulations on brain activity in embryo.

## Editor's evaluation

This study represents valuable findings of asymmetric brain development. The data, collected by applying MEMRI in chick embryos, were considered solid. While further studies are needed to examine what causes asymmetric development, the findings reported here will open new pathways for studying experience-dependent brain development.

**\*For correspondence:**
elena.lorenzi@unitn.it

**Competing interest:** The authors declare that no competing interests exist.

## Introduction

How the brain develops left-right asymmetry is poorly understood. Some of the molecular cues that specify the left-right axis of the body are guided by genes of the Nodal cascade (*Halpern et al., 2003*; *Rogers et al., 2013*; *Ocklenburg and Güntürkün, 2017*). These genes are involved in the establishment of the asymmetric morphology and positioning of the visceral organs (*Ramsdell and Yost, 1998*), but also produce a slight torsion of the embryo with the forehead pointing to the right. This rightward spinal torsion seems to occur in all amniotes (*Zhu et al., 1999*) including human embryos (*Ververs and Hopkins, 1994*).

The torsion is particularly prominent in birds. In most species, the head of the embryo is turned to the side, so that the left eye is completely occluded by the body, while the right eye faces the eggshell (see *Figure 1a*). Thus, light stimulation passing through the eggshell and membranes reaches mostly the right eye.

Fibers coming from each eye decussate almost completely at the level of the optic chiasm. In the thalamofugal pathway, they then ascend contralaterally to the thalamus (more precisely at the level of the nucleus geniculatus lateralis pars dorsalis, Gld) and from there mostly ipsilaterally to the visual Wulst (an avian equivalent of the primary visual cortex of mammals). In chicks it has been shown that a small portion of visual fibers, so called supraoptic decussation, re-cross from the thalamus (*Figure 1a*) and innervate the contralateral side of the Wulst. Following light stimulation in embryo during a

sensitive period embryonic days E17–18 (*Rogers and Deng, 1999*), however, in previous experiments it has been shown that light stimulation in the embryo result, on post-hatching day 2, in more projections from the left thalamus (receiving visual input from the right eye) than from the right thalamus to the right Wulst (*Rogers and Deng, 1999*) (see *Figure 1a* for an anatomical reference). Here, we attempted to image in vivo the embryonic functional mechanisms underlying the establishment of such an asymmetry.

We used manganese enhanced magnetic resonance imaging (MEMRI) to visualize embryonic brain activity during the sensitive window inside the eggs of domestic chicks (*Gallus gallus*). Paramagnetic manganese ions $Mn^{2+}$ enter excitable cells via voltage-gated calcium channels, and can be transported along axons and across synapses (*Massaad and Pautler, 2011*). Hence, changes in the regional contrast of magnetic resonance images induced by activity-dependent accumulation of manganese provide a means to map neuronal activation elicited by stimuli (*Lin and Koretsky, 1997*) and to trace the axonal tracts involved (*Pautler et al., 1998*; *Van der Linden et al., 2007*). On E17 we administered manganese to the embryos via the chorioallantoic membrane. Immediately after, one group of embryos was incubated with light (light) and the other was incubated in the darkness (dark). On E18 eggs were placed in the scanner (MR) and functional T1 maps were acquired.

## Results

The ANOVA revealed a significant main effect of brain region ($F_{(2,32)}$ = 3.693, p=0.036) and a significant interaction between brain region and condition ($F_{(2,32)}$ = 4.274, p=0.023), while no significant main effect of condition ($F_{(1,16)}$ = 0.335, p=0.571).

A significant difference between conditions was found in the thalamus ($t_{(16)}$ = -2.710, pFDR = 0.045, d = -1.286), which showed higher manganese uptake/activity in the dark condition (dark: mean -0.038 ± s.e.m. 0.006; light: -0.015±0.006). No significant differences were found between conditions in either the visual Wulst ($t_{(16)}$ = 1.280, pFDR = 0.329, d=0.607; dark: -0.007±0.005; light: -0.020±0.009) or the optic tectum ($t_{(16)}$ = -0.148, pFDR = 0.884, d = -0.070; dark: -0.010±0.009; light: -0.009±0.006).

A significant right lateralization was detected in the thalamus in both groups (dark: $t_{(7)}$ = -6.090, p≤0.001 (uncorrected), d = -2.153; light: $t_{(9)}$ = -2.419, p=0.039 (uncorr.), d = -0.765). The visual Wulst was significantly right lateralized in the light group ($t_{(9)}$ = -2.383, p=0.041 (uncorr.), d = -0.754), but no significant lateralization emerged in the dark group ($t_{(7)}$ = -1.350, p=0.219, d = -0.477). No significant lateralization was found in the optic tectum either of the groups (dark: $t_{(7)}$ = -1.149, p=0.288 (uncorr.), d = -0.406; light: $t_{(9)}$ = -1.493, p=0.170 (uncorr.), d = -0.472).

All data generated or analyzed during this study are included in the manuscript and supporting file; Source Data files have been provided for *Figure 1c* (https://doi.org/doi:10.5061/dryad.9cnp5hqp2).

## Discussion

We detected a spontaneous (dark) as well as an induced (light) lateralization of brain activity in the thalamofugal visual pathway (thalamus); in contrast, no lateralization nor any effect of light was observed in the optic tectum (see *Figure 1c*). After light stimulation the manganese accumulation revealed a right lateralized brain activity in the Wulst, congruent with increased activity from the left thalamus to the right Wulst (via the supraoptic decussation). Interestingly, however, in the complete absence of visual stimulation (dark), we found a strong right lateralization in the thalamus; light exposure seemed to reduce such spontaneous asymmetry, as a result of light stimulation to the right eye that likely increased activity in the left thalamus (due to complete decussation at the level of optic chiasm). Thus, manganese accumulation in the tissue seems to be directly linked to the promotion of the growth of the fibers along the supraoptic decussation from the left thalamus to the right Wulst.

Asymmetry in manganese accumulation can be interpreted as a result of prolonged neuronal stimulation under illumination compared to the dark condition. Changes in cellular density may also contribute to differential distribution of Mn. However, within the temporal frame of the experiment, we deem it unlikely that altered cellularity is the main mechanism, as we should assume several-fold changes in cell density to explain the observed differences in T1. Similarly, light-induced region-specific changes in blood brain barrier permeability appear unlikely under the experimental conditions of this study.

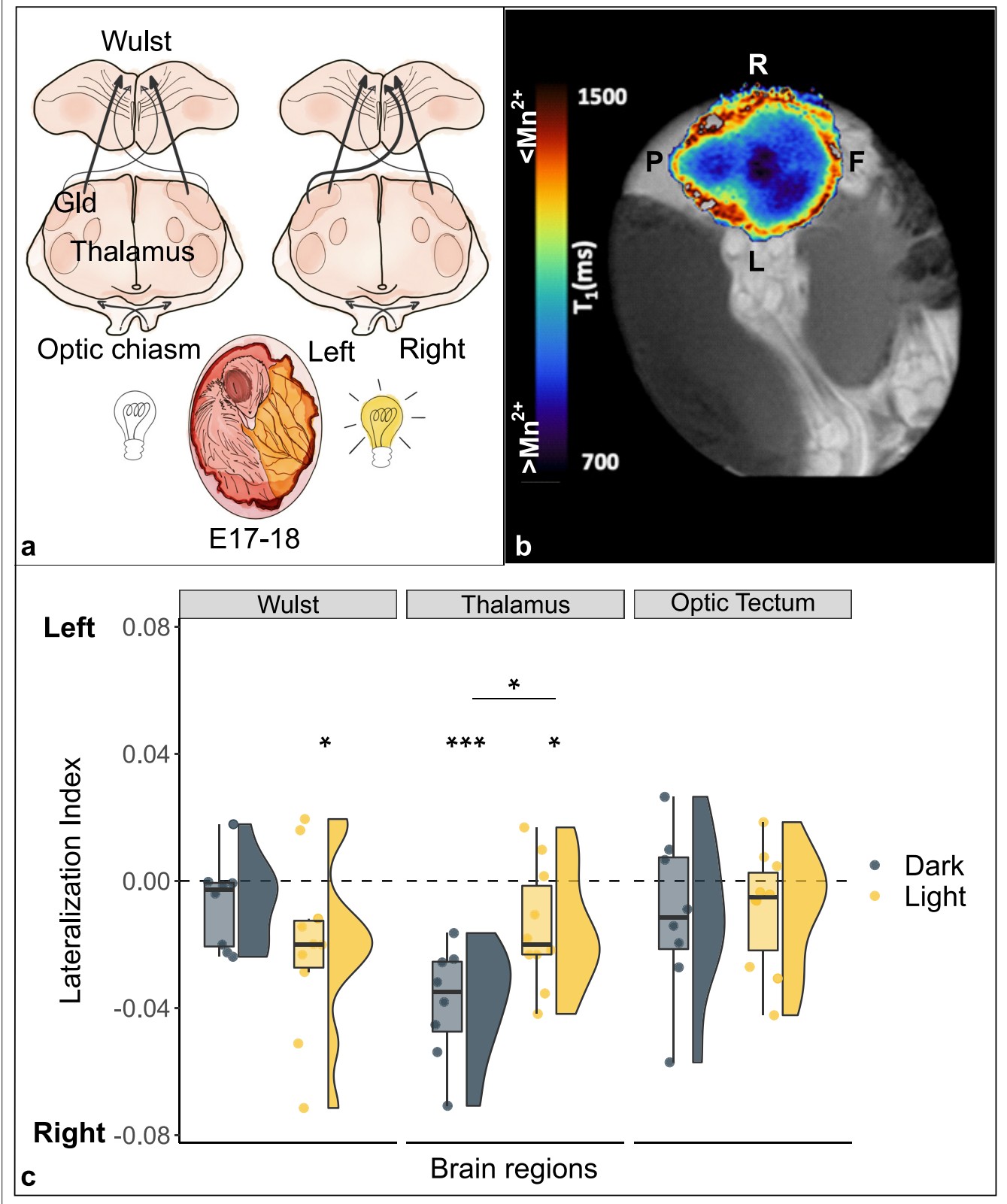

**Figure 1.** Lateralization of brain activity in chick embryos light and dark incubated. (**a**) Schematic representation of the impact of light (off/on) on the lateralization of projections from the left thalamus (nucleus geniculatus lateralis pars dorsalis, Gld) to the right Wulst during the sensitive window (E17–18). (**b**) Typical longitudinal relaxation time (**T1**) map of a chick's brain after administration of manganese. $Mn^{2+}$ is a paramagnetic ion and shortens the T1 of nuclear spins in brain tissue in a concentration-dependent manner. Blue-darker regions correspond to higher intracellular $Mn^{2+}$ accumulation,

*Figure 1 continued on next page*

*Figure 1 continued*

and hence to increased neuronal activity. Left-right front-posterior coordinates (**L, R, F, P**) are indicated with respect to the sagittal brain axis of embryo chick brain inside the egg. (**c**) Brain activity is reported as lateralization index (1=activity only in the left hemisphere; –1=only in the right). Regions of the thalamofugal pathway: Wulst and thalamus (Gld); region of the tectofugal pathway: optic tectum. Asterisk between groups indicates a significant difference between experimental conditions (Dark n=8, Light n=10). Asterisk above single group indicates lateralization significantly different from chance (0, dotted line). * indicates p<0.05, *** indicates p≤0.001. To best represent data, we used a raincloud plot that combines for each region and for each group: boxplot, raw data (circles) and data distribution (half violin on the left).

The online version of this article includes the following figure supplement(s) for figure 1:

**Figure supplement 1.** T1w structural image of a chick embryo.

**Figure supplement 2.** Representative in ovo MRI of a full egg.

**Figure supplement 3.** Visual assessment of two representative voxelwise T1 maps from a dark-incubated (left) and light-incubated egg (right).

**Figure supplement 4.** Saturation recovery curve of inter-hemispheric signal intensity of the thalamus.

Light stimulation during the sensitive window appears to be crucial in establishing brain asymmetry at the onset of life. However, spontaneous lateralization of brain activity seems to be already there during embryonic development in the complete absence of external light stimulation (for spontaneous lateralized activity soon after hatching, see *Lorenzi et al., 2019*). It has been hypothesized that a similar interplay between a genetically determined rightward torsion of the body and the resulting asymmetry in sensory stimulation (even for other sensory modalities) may represent a general mechanism for brain asymmetry in all vertebrates, including humans (*Rogers et al., 2013*).

This new protocol of MEMRI allowed us for the first time to map in vivo in the embryo the lateralized pattern of brain activity at the early stages of brain development. The MEMRI protocol here established could be further exploited to investigate early embryonic learning that occurs as a result of sensory stimulation through the egg before birth.

## Materials and methods

### Subjects and embryonic manipulations

Fresh fertilized eggs of the Aviagen Ross 308 strain (*Gallus gallus domesticus*; provider: Azienda Agricola Crescenti Brescia, Italy) were incubated in the laboratory under constant temperature and humidity in the complete darkness (37.7°C; 60%; incubator: FIEM MG140/200 Rural LCD EVO). On embryonic day 17 (E17), the eggshell was pierced in correspondence of the air sac. 80 µl of manganese chloride (540 mM $MnCl_2$ tetrahydrate, Sigma-Aldrich) dissolved in phosphate-buffered saline (0.1 M, pH = 7.4, 0.9% sodium chloride) were dropped on the chorioallantoic membrane. Paper tape was used to seal the eggshell again and eggs were placed back in the incubator. After manganese injection a group of 10 eggs were exposed to 17.30 hr of light (light group), by mean of a rectangular plastic translucent panel (36×38 cm$^2$), with 15 LEDs (270 lm) homogeneously distributed on it placed on the roof of the incubator 15 cm above the eggs. Another group of eight eggs were incubated in the darkness (dark group). On E18 each egg was taken individually and randomly from one of the two incubators (light or dark), placed inside a dark box and moved to the fridge (4°C) for 30 min in order to avoid any movement during scanning. Sample size was determined based on previous experiments on newly hatched chicks (*Halpern et al., 2003*).

### Image acquisition

Images were acquired at 7 T on a Bruker Pharmascan (https://www.bruker.com) with a 72 mm i.d. single-channel transmitter-receiver birdcage resonator. Pulse sequence parameters for RARE (rapid acquisition with relaxation enhancement) variable TR T1 mapping method (RARE-VTR) acquisition were set as follows: TR =[169.3; 240.6; 340.6; 440.6; 540.6; 640.6; 740.6; 840.6; 990.6; 1490.6; 1990.6; 2990.6; 4990.6] ms, TE = 6.25 ms, MTX = 200 × 200 × 9, voxel size = 0.25 × 0.25 × 1 mm, ACQtime = 18 min 30 s. In addition, a series of multislice T1W FLASH (fast low angle shot) images (TR = 91 ms, TE = 4 ms, MTX = 200 × 200 × 7, voxel size = 0.275 × 0.275 × 12 mm, ACQtime = 13.6 s) were acquired for anatomical reference and to define the optimal imaging plane for RARE-VTR. To avoid artifacts caused by spontaneous movement of the chick embryo, eggs were kept at 4°C in a temperature-controlled refrigerator for 20 min prior to the acquisition. This ensured the immobilization of the chick embryo

within the egg due to mild hypothermia for about 30/45 min. In this time window, the egg was placed in a foam pad enclosure to further attenuate mechanical vibrations induced by imaging gradient switching and inserted into the birdcage coil for MR scanning procedures.

## Image analyses

Regions of interest (ROIs) were selected from the two main visual pathways, the thalamo- and the tectofugal pathway blinded from the experimental condition. For the thalamofugal visual pathway a portion of the dorsolateral thalamus and of the visual Wulst were selected, of which respectively three and two sections per each hemisphere were quantified. For the tectofugal visual pathway only the optic tectum was selected, of which four sections per each hemisphere were quantified. ROIs were segmented manually on the first volume of the RARE-VTR dataset of each animal using the free software ITK-SNAP (https://www.itksnap.org/; *Yushkevich et al., 2006*). Quantitative T1 maps were obtained by fitting the general signal recovery model in Matlab (https://www.mathworks.com) to produce voxelwise distributions of T1 relaxation time over the whole brain.

## Data analyses

In order to investigate region-specific increase in manganese uptake into the brain and therefore suggesting a sustained and prolonged activation, average T1 values were computed via ROI analysis and transformed using the formula:

$$t1 = \left(\frac{1}{T1}\right) \times 1000$$

The dependent variable employed in all the analyses was a left lateralization index computed for each brain region using the formula:

$$\text{Lateralization index} = \frac{\text{Left t1} - \text{Right t1}}{\text{Left t1} + \text{Right t1}}$$

Lateralization index values ranged from 1 (brain activity present only in the left hemisphere) to – 1 (brain activity present only in the right hemisphere).

The difference in the lateralization between the two experimental conditions, for the three brain regions, was tested with a repeated measurement ANOVA, between-subjects factor 'condition' (two levels: light, dark), within-subject factor 'brain region' (three levels: Wulst, thalamus, optic tectum). False discovery rate (FDR) was applied to correct for multiple comparisons. For post hoc analyses, one-sample two-tailed t-tests were used to compare lateralization indexes against the expected chance level (0). Two-tailed independent samples t-tests were used to compare indexes between conditions. For every t-test the Cohen's d was calculated as a measure of the effect size. No correction for multiple comparisons was applied to the post hoc analyses. All the analyses and the raincloud plot were performed using the software R (https://www.r-project.org/).

# Acknowledgements

GV acknowledges support from the European Research Council (ERC) under the European Union's Horizon 2020 research and innovation program (grant agreement SPANUMBRA No. 833504).

# Additional information

## Funding

| Funder | Grant reference number | Author |
|---|---|---|
| European Research Council | 833504 | Giorgio Vallortigara |

The funders had no role in study design, data collection and interpretation, or the decision to submit the work for publication.

## Author contributions
Elena Lorenzi, Conceptualization, Investigation, Methodology, Writing – original draft, Project administration; Stefano Tambalo, Data curation, Software, Formal analysis, Investigation, Visualization, Writing – original draft, Project administration; Giorgio Vallortigara, Angelo Bifone, Conceptualization, Resources, Supervision, Funding acquisition, Writing – review and editing

## Author ORCIDs
Elena Lorenzi https://orcid.org/0000-0001-8670-4751
Stefano Tambalo http://orcid.org/0000-0003-2562-1324
Giorgio Vallortigara http://orcid.org/0000-0001-8192-9062

## Decision letter and Author response
Decision letter https://doi.org/10.7554/eLife.86116.sa1
Author response https://doi.org/10.7554/eLife.86116.sa2

# Additional files

## Supplementary files
• Supplementary file 1. Region of interest (ROI)-averaged T1 values measured for the two experimental groups. All values are expressed in milliseconds.
• MDAR checklist

## Data availability
All data generated or analysed during this study are included in the manuscript and supporting file; Source Data files are freely available here: https://doi.org/10.5061/dryad.9cnp5hqp2.

The following dataset was generated:

| Author(s) | Year | Dataset title | Dataset URL | Database and Identifier |
|---|---|---|---|---|
| Lorenzi E, Tambalo S, Vallortigara G, Vallortigara G, Bifone A | 2023 | Manganese Enhanced Magnetic Resonance Imaging reveals light-induced brain asymmetry in embryo | https://doi.org/10.5061/dryad.9cnp5hqp2 | Dryad Digital Repository, 10.5061/dryad.9cnp5hqp2 |

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
