## [Editor Report]

This study represents valuable findings of asymmetric brain development. The data, collected by applying MEMRI in chick embryos, were considered solid. While further studies are needed to examine what causes asymmetric development, the findings reported here will open new pathways for studying experience-dependent brain development.

---

## [Decision Letter]

**Decision letter after peer review:**

Thank you for submitting your article "Manganese Enhanced Magnetic Resonance Imaging reveals light-induced brain asymmetry in embryo" for consideration by *eLife*. Your article has been reviewed by 3 peer reviewers, and the evaluation has been overseen by a Reviewing Editor and Christian Rutz as the Senior Editor. The following individuals involved in review of your submission have agreed to reveal their identity: Lesley J. Rogers (Reviewer #1); Sanne Moorman (Reviewer #3).

Essential revisions:

Reviewers found difficulties to assess their findings as enough data was not provided. We require to revise the manuscript fully with more detailed data and methodological proof for the current finding.

1) Please provide MRI anatomical scan image for showing the definition of the brain region

2) Please also provide images to support your findings, such as light sensitivity, laterality etc.

*Reviewer #1 (Recommendations for the authors):*

28: I would say in most species, rather than several. Although I recognise that there is only empirical evidence for several species, it is likely to be only megapodes that do not turn the head so that the left eye is occluded by the embryo's body.

35-40: I think the reader will be confused by which side is contralateral and to what. Perhaps it would be better to say contralateral or ipsilateral to the eye sending visual inputs to the brain. Also, in line 38, it is said "there seems to be...", whereas this is a fact. Replace 'seems to be' by 'are' and make it clear that this asymmetry is present after light stimulation and clearly evident on day 2 post-hatching (Rogers and Deng (1999) Behavi. Brain Res. 98, 277-287).

52: The calculation of the lateralization index needs to be explained clearly.

Figure 1b: This figure is not explained well. Shouldn't left and right be on the left and right sides instead of above and below? Also, the footnote says, "Darker regions correspond to higher intracellular Mn^2+^ accumulation" but the photograph is colour coded. What does darker mean in this case?

Figure 1c: The footnote does not say exactly what is plotted: e.g., median values, and what indicates the data spread? What are the distributions plotted to the right of each median etc and are they necessary? It would be clearer without them.

64, footnote to Figure 1: There are no asterisks below the groups. They are above them.

81-82: Correct 'of both groups' to 'either of the groups'.

119: Change 'to scrutiny' to 'to scrutinise'.

*Reviewer #2 (Recommendations for the authors):*

1. The most important issue for this reviewer is the absence of an anatomical scan (a high resolution T2 scan or anything else). This will help in brain region delineation. The study is done in an embryo using MRI which makes the anatomical scans even more necessary.

2. The results also accompany images taken only at dark and only at light. This will help the readers and reviewers to see the MnCl2 movement through brain regions in dark (absence of sensory stimulation). Further, images need to be taken just before and after MnCl2 injections. This will help in determining if equal amount of MnCl2 was injected in light and dark group. Although T1 images are good for making absolute MnCl2 quantifications, we don't know if uptake/transport of MnCl2 through brain regions is concentration dependent. Thus equal amount is necessary to compare between groups. Also I would appreciate would images obtained after subtraction of dark and light group brains to see the differences/areas.

3. The study claims to understand light induced brain development/lateralization by using MnCl2 as a contrast agent. Brain lateralization of chicks inside the egg depends on the side of the embryo which is facing the light inducing more development in one side. However, MnCl2 will be taken up and move more as the areas, organ function. For example eye. Thus, the results may also be influenced by eye's functioning. So, MnCl2 quantification should have done in eyes too to test if both eyes have comparable functioning.

4. Selection of regions from tectofugal and thamalmo fugal pathways: Wulst is the farthest. Tectum is the closest. Even though lateralization is tested, measurements should be made for all areas. Consider this scenario: suppose there is no lateralization in Wulst but a strong one in Gld. The MnCl2 transfers from Gld to Wulst. So the Wulst will inherently show lateralized effect of light.

5. In connection to the above point, images taken at regular intervals to show the transport of MnCl2 from site of injection to end area would have been nice. We have done similar studies in adult birds and this is absolutely necessary to study the visual system. Also this will help in measuring the flow of MnCl2.

6. A very important consideration to be taken in account is the presence of deep brain photoreceptors in avian brain. Light may influence the MnCl2 contrast through these receptors which will not necessarily support lateralization or brain development. Thus, images only in light needs to be shown for readers to appreciate lights affect on brain MnCl2 contrasts.

7. I fail to understand how strong Lateralization of right thalamus becomes 'less' Lateralization on light stimulation (Figure 1).

8. As a fellow MRI researcher, I would highly appreciate seeing T1 maps and other standardizations in the supplementary data. MRI of bird brains is uniquely difficult and MRI of embryo brain inside eggs even more so.

*Reviewer #3 (Recommendations for the authors):*

– Manganese ions can indicate both neural activity and axonal connectivity. Asymmetric connectivity is considered a main cause for lateralized activation patterns in the literature. I think you should also report if you found any differences in lateralized connectivity between the light and dark groups. It is important to learn about the strengths of contralateral projections in the light vs dark groups for answering the research question, and the MEMRI technique allows for including this analysis if I understand it correctly.

– It is not clear to me what the image in Figure 1B shows: the legend states it is an embryonic pigeon brain, but is it showing the whole egg? What is the size of the manganese positive area in relation to the size of the brain? The image indicates L-R but I cannot recognise front-back etc.

– It feels like I cannot judge the strengths and weaknesses of the main data in Figure 1C. The authors should share more MRI images as a visual example of manganese levels in the different brain regions between the two groups.

– Results show that the thalamus was more active in dark than light group. It might be interesting to discuss what kind of role the thalamus has in the visual pathway. For example, does it mainly have inhibitory connections, where less activity with light-stimulation leads to disinhibition of downstream areas?

– I think that the interpretation that more bilateral thalamic activation leads to more right-dominant Wulst activity is not clear. I would assume that the contralateral projections would be smaller than the ipsilateral connections. If this is true, then more left-sided thalamic activity in the light (than dark) group should drive more activity in the left-sided Wulst (while at the same time also the right-sided Wulst, but not to the extent of net right-dominance).

– Neural activity in the nucleus rotundus and/or entopallium should be included in the analyses to be able to discuss lateralization of the tectofugal pathway.

– It is not clear from the introduction or figure 1A what pathways are measured and compared. Figure 1A should show the two pathways instead of only the thalamofugal one, or another figure could be added for the tectofugal pathway.

– With the post-hoc t-testing, the authors should correct for multiple testing.

– Results should be compared to relevant literature, such as the following:

Rogers, 2006, Cortex, doi: 10.1016/S0010-9452(08)70332-0

Valencia-Alfonso et al., 2009, Phil. Trans. R. Soc. B, doi:10.1098/rstb.2008.0240

Concha et al., 2012, Nat Rev Neurosci, doi:10.1038/nrn3371

Manns and Ströckens, 2014, Frontiers Psych, doi: 10.3389/fpsyg.2014.00206

– It is my understanding that projections from optical tectum to thalamus are symmetric if eggs are incubated in the dark (Gunturkun 1993; Rogers, 2006). This result from the literature should be cited, and it should be discussed how the current results relate to it.

---

## [Author Response]

Essential revisions:Reviewers found difficulties to assess their findings as enough data was not provided. We require to revise the manuscript fully with more detailed data and methodological proof for the current finding.1) Please provide MRI anatomical scan image for showing the definition of the brain region2) Please also provide images to support your findings, such as light sensitivity, laterality etc.

We value the feedback provided by the Editor and Reviewers and recognize the significance of addressing their concerns. In response, we have implemented several amendments to the manuscript. A summary of the revisions made is listed below:

1a) MRI anatomical scan image: We included in the Supplementary Information a high-resolution 3D MRI anatomical scan image (Figure supplement 1) that clearly displays the in-ovo localization of the chick brain.

1b) Image set for Supplementary Information section: A comprehensive image grid for the SI, featuring an anatomical scan labeled with relevant Regions Of Interest (ROIs), has been added (Figure supplement 2).

1c) Table with ROIs: Supplementary Information section now includes a table (Table S1) containing raw data from the list of ROIs relevant to our study. We ensure that all essential ROIs, including the eye region, are reported.

2) To further complement our findings, we have added Figure supplement 3 and 4 in the Supplementary Information section. These two graphical elements show, respectively: (i) a side-to-side comparison of representative voxelwise T1 maps calculated in the two experimental conditions; (ii) Group-averaged T1 saturation recovery curve extracted from the Thalamus. A differential uptake of MnCl_2_ is derived by the condition-dependent modulation of signal intensity over time.

By implementing these revisions, we aim to provide a more clear and comprehensive understanding of our work. Given the restrictions of a Brief Communication (only one figure allowed in the main body of the manuscript), the data have been added to the Supplementary Information section.

Reviewer #1 (Recommendations for the authors):28: I would say in most species, rather than several. Although I recognise that there is only empirical evidence for several species, it is likely to be only megapodes that do not turn the head so that the left eye is occluded by the embryo's body.35-40: I think the reader will be confused by which side is contralateral and to what. Perhaps it would be better to say contralateral or ipsilateral to the eye sending visual inputs to the brain. Also, in line 38, it is said "there seems to be...", whereas this is a fact. Replace 'seems to be' by 'are' and make it clear that this asymmetry is present after light stimulation and clearly evident on day 2 post-hatching (Rogers and Deng (1999) Behavi. Brain Res. 98, 277-287).

We thank the reviewer for pointing this out, we have revised the manuscript accordingly: l. 48 we substituted “several” for “most” as suggested; ll. 59-61 we stated it more clearly “Following light stimulation in embryo during a sensitive period (embryonic days E17-18^7^), however, on post-hatching day 2 there are more projections from the left (receiving visual input from the right eye) than from the right thalamus to the right Wulst^7^ (see Figure 1a).”. Moreover, we have also added the paper suggested by the Reviewer to the bibliography.

52: The calculation of the lateralization index needs to be explained clearly.

The definition of the lateralization index has been clarified in the Supplementary Information (in the Data analyses section ll. 249-251).

Figure 1b: This figure is not explained well. Shouldn't left and right be on the left and right sides instead of above and below? Also, the footnote says, "Darker regions correspond to higher intracellular Mn^2+^ accumulation" but the photograph is colour coded. What does darker mean in this case?

We thank the reviewer for pointing this out. We have used the frame of reference of the brain coordinates (front-post-left-right) with respect to the sagittal axis of the brain inside the egg. We have now clarified this point in the caption to avoid any confusion (ll. 83-85). We have also modified the image adding “F” for frontal and “P” for posterior to better exemplify the reference coordinates.

Figure 1c: The footnote does not say exactly what is plotted: e.g., median values, and what indicates the data spread? What are the distributions plotted to the right of each median etc and are they necessary? It would be clearer without them.

The graph of Figure 1c reports the lateralization index as defined in the data analysis section. We decided to represent data using raincloud plots (Allen, M., Poggiali, D., Whitaker, K., Marshall, T. R., and Kievit, R. A. (2019). Raincloud plots: a multi-platform tool for robust data visualization. Wellcome open research, 4.), that combine standard boxplots, with jittered raw data and violin plots for data distribution frequency. We explained this kind of plot in deeper detail in the figure caption (ll. 89-91).

64, footnote to Figure 1: There are no asterisks below the groups. They are above them.

The caption has been amended.

81-82: Correct 'of both groups' to 'either of the groups'.119: Change 'to scrutiny' to 'to scrutinise'.

We thank the reviewer for noticing these, we have revised the manuscript accordingly to the suggestions.

Reviewer #2 (Recommendations for the authors):1. The most important issue for this reviewer is the absence of an anatomical scan (a high resolution T2 scan or anything else). This will help in brain region delineation. The study is done in an embryo using MRI which makes the anatomical scans even more necessary.

We have added anatomical reference images and the definition of the ROIs in the Supplementary Information section (the Short Report only allows one image in the main body of the manuscript).

Specifically:

1a) MRI anatomical scan image: We included in the Supplementary Information a high-resolution 3D MRI anatomical scan image (Figure supplement 1) that clearly displays the in-ovo localization of the chick brain.

1b) Image set for Supplementary Information section: A comprehensive image grid for the Supplementary Information, featuring an anatomical scan labeled with relevant Regions Of Interest (ROIs), has been added (Figure supplement 2).

1c) Table with ROIs: Supplementary Information section now includes a table (Table S1) containing raw data from the list of ROIs relevant to our study. We ensure that all essential ROIs, including the eye region, are reported.

2) To further complement our findings, we have added Figure supplement 3 and 4 in the Supplementary Information section. These two graphical elements show, respectively: (i) a side-to-side comparison of representative voxelwise T1 maps calculated in the two experimental conditions; (ii) Group-averaged T1 saturation recovery curve extracted from the Thalamus. A differential uptake of MnCl_2_ is derived by the condition-dependent modulation of signal intensity over time.

2. The results also accompany images taken only at dark and only at light. This will help the readers and reviewers to see the MnCl2 movement through brain regions in dark (absence of sensory stimulation). Further, images need to be taken just before and after MnCl2 injections. This will help in determining if equal amount of MnCl2 was injected in light and dark group. Although T1 images are good for making absolute MnCl2 quantifications, we don't know if uptake/transport of MnCl2 through brain regions is concentration dependent. Thus equal amount is necessary to compare between groups. Also I would appreciate would images obtained after subtraction of dark and light group brains to see the differences/areas.

We have added a few examples of T1 maps for subjects exposed to light and kept in the dark, for the reader’s perusal.

The use of MnCl_2_ as a long-term marker of brain activity is grounded on a particularly slow pharmacokinetics and an extremely long washout curve. This is primarily attributed to the process of Mn^2+^ ion accumulation within postsynaptic neurons, where it competes with ca^2+^ through calcium channels. This accumulation occurs in an activity-dependent manner over a significant time span (hours), enabling the capture of prolonged and chronic stimuli, as reported in this study. Moreover, in our model we used systemic injection of MnCl_2_, thus extending the kinetics of contrast agent accumulation in the brain tissue. Considering these factors, pre- and post-administration images would predominantly display the accumulation at the injection site, rather than providing informative details regarding concentration and tissue distribution.

To control for unbalanced administration of contrast agent, the injected volume is determined to ensure an equal amount of manganese concentration per unit of egg-weight.

Regarding the subtraction of light/dark images, conducting paired subtractions across different individuals in this cross-sectional design would likely introduce undesired biases to the analysis. To mitigate this problem, we opted for an ROI-based approach to investigate group-wise differences in MnCl_2_ accumulation.

Additionally, we have also reported the saturation recovery curves in the SI section.

3. The study claims to understand light induced brain development/lateralization by using MnCl2 as a contrast agent. Brain lateralization of chicks inside the egg depends on the side of the embryo which is facing the light inducing more development in one side. However, MnCl2 will be taken up and move more as the areas, organ function. For example eye. Thus, the results may also be influenced by eye's functioning. So, MnCl2 quantification should have done in eyes too to test if both eyes have comparable functioning.

We have added a table in the Supplementary Information section with the Mn uptake by the eye, for completeness. No significant difference between eyes was observed.

4. Selection of regions from tectofugal and thamalmo fugal pathways: wulst is the farthest. Tectum is the closest. Even though lateralization is tested, measurements should be made for all areas. Consider this scenario: suppose there is no lateralization in wulst but a strong one in Gld. The MnCl2 transfers from Gld to wulst. So the wulst will inherently show lateralized effect of light.

To the best of our knowledge optic tectum in chicks does not send ascending projections to the visual Wulst (Deng, C., and Rogers, L. J. (1998). Bilaterally projecting neurons in the two visual pathways of chicks. Brain Research, 794(2), 281-290.). Visual Wulst receives visual input from Gld in the thalamus via the dorsal supraoptic decussation and intra-hemispheric projections.

From a methodological perspective, we note that MnCl_2_ was administered systemically through the CAM. This results in a distribution of the Mn ions in the brain parenchyma, since the Blood Brain Barrier at this embryonic stage is not completely developed and its permeability to ions and small molecules is way higher in embryo than at later stages of development (Engelhardt, B. (2003). Development of the blood-brain barrier. Cell and tissue research, 314(1), 119-129.). Therefore, the distribution of manganese in the brain is relatively uniform, and its accumulation likely reflects local neuronal activity, rather than trans-synaptic transfer from upstream neurons.

The scenario described by the Reviewer is an interesting one, and could be explored with a different protocol involving local administration of MnCl_2_ in specific brain regions, as in tract tracing studies. In this case, the downstream accumulation of Mn would be weighted by upstream activity, and by interneuron synaptic activity.

While this is definitely worth investigating in future studies, in this Short Communication we intended to share our findings in a brief form and at an early stage, thus enabling the use of this novel and promising protocol by the scientific community involved in the investigation of embryonic neurodevelopment.

5. In connection to the above point, images taken at regular intervals to show the transport of MnCl2 from site of injection to end area would have been nice. We have done similar studies in adult birds and this is absolutely necessary to study the visual system. Also this will help in measuring the flow of MnCl2.

As noted above, here we used systemic administration of manganese, and the build-up of its accumulation is the combined result of the kinetics of the contrast agent and of its activity dependent accumulation. By manipulating exposure to light in two parallel groups, we studied the time-integrated effects of sensory stimulation.

Conversely, in tract tracing studies, Mn is injected locally in the brain parenchyma, and the distal transport of manganese reflects active transport only. The suggestion by the reviewer is an extremely interesting one, and will be the subject of future investigations. We have also added a reference pointing the reader to alternative MEMRI protocols [Van der Linden et al., 2007: l. 71].

6. A very important consideration to be taken in account is the presence of deep brain photoreceptors in avian brain. Light may influence the MnCl2 contrast through these receptors which will not necessarily support lateralization or brain development. Thus, images only in light needs to be shown for readers to appreciate lights affect on brain MnCl2 contrasts.

We thank the Reviewer for this suggestion. However, we observed lateralization in the hemisphere that is farther away from the light source (left hemisphere; Figure 1b). This evidence is not consistent with the hypothesis that asymmetry observed is the result of direct activation of deep-brain photoreceptors, that should be more stimulated in the hemisphere closer to the light source (right hemisphere).

7. I fail to understand how strong Lateralization of right thalamus becomes 'less' Lateralization on light stimulation (Figure 1).

We observed spontaneous strong lateralization in the right thalamus of embryos incubated in the darkness, which differed significantly from embryos incubated with light. Specifically, we measured more symmetrical distributions on Mn in the thalamus for the light exposed group than for the dark group. Light stimulation at this embryonic stage is reaching only the right eye, and visual input decussates at the level of the optic chiasm reaching the left thalamus. Therefore, we hypothesize that light stimulation is increasing brain activity in the left thalamus, thus decreasing the spontaneous level of asymmetry in this region. We have clarified this point in the revised manuscript (ll. 124-126).

8. As a fellow MRI researcher, I would highly appreciate seeing T1 maps and other standardizations in the supplementary data. MRI of bird brains is uniquely difficult and MRI of embryo brain inside eggs even more so.

T1 maps and reference images have been included in the Supplementary Information section. Moreover, we have included the saturation recovery curves, to enable the reader to directly inspect the acquired data before the T1 is extracted by curve fitting.

Reviewer #3 (Recommendations for the authors):– Manganese ions can indicate both neural activity and axonal connectivity. Asymmetric connectivity is considered a main cause for lateralized activation patterns in the literature. I think you should also report if you found any differences in lateralized connectivity between the light and dark groups. It is important to learn about the strengths of contralateral projections in the light vs dark groups for answering the research question, and the MEMRI technique allows for including this analysis if I understand it correctly.

Here we used systemic administration of Mn. The Blood Brain Barrier at this embryonic stage is not completely developed and that its permeability to ions and small molecules is way higher in embryo than in later stages of development (Engelhardt, B. (2003). Development of the blood-brain barrier. Cell and tissue research, 314(1), 119-129.). Other studies involving direct, local injection in selected brain regions are more apt to investigate connectivity, but this is not the protocol used here. We appreciate the reviewer’s suggestion, and this will be the object of future experiments. However, we would like to disseminate the current protocol and the results it led to at an early stage to enable and encourage other researchers in the field.

– It is not clear to me what the image in Figure 1B shows: the legend states it is an embryonic pigeon brain, but is it showing the whole egg? What is the size of the manganese positive area in relation to the size of the brain? The image indicates L-R but I cannot recognise front-back etc.

Figure 1b shows the entire domestic chick egg (*Gallus gallus domesticus*), we have used the frame of reference of the brain coordinates (front-post-left-right) with respect to the sagittal axis of the brain inside the egg. We have now clarified this point in the caption to avoid any confusion (ll. 84-85). We have also modified the image adding “F” for frontal and “P” for posterior to better exemplify the reference coordinates. Regarding the question about the size of the manganese-positive area relative to the brain size, we would like to provide clarification. MnCl_2_ diffuses throughout the entire brain tissue, resulting in regional differences primarily influenced by activity-dependent uptake. This uptake is proportional to the time integral of sustained activity within each specific region. Consequently, it is not possible to define a distinct "manganese-positive" area, as the MnCl_2_ permeates the entire parenchyma of the brain.

– It feels like I cannot judge the strengths and weaknesses of the main data in Figure 1C. The authors should share more MRI images as a visual example of manganese levels in the different brain regions between the two groups.

We have added substantial more data, including T1 maps and reference images, in the Supplementary Information section.

Specifically:

1a) MRI anatomical scan image: We included in the Supplementary Information a high-resolution 3D MRI anatomical scan image (Figure supplement 1) that clearly displays the in-ovo localization of the chick brain.

1b) Image set for Supplementary Information section: A comprehensive image grid for the Supplementary Information, featuring an anatomical scan labeled with relevant Regions Of Interest (ROIs), has been added (Figure supplement 2).

1c) Table with ROIs: Supplementary Information section now includes a table (Table S1) containing raw data from the list of ROIs relevant to our study. We ensure that all essential ROIs, including the eye region, are reported.

2) To further complement our findings, we have added Figure supplement 3 and 4 in the Supplementary Information section. These two graphical elements show, respectively: (i) a side-to-side comparison of representative voxelwise T1 maps calculated in the two experimental conditions; (ii) Group-averaged T1 saturation recovery curve extracted from the Thalamus. A differential uptake of MnCl_2_ is derived by the condition-dependent modulation of signal intensity over time.

– Results show that the thalamus was more active in dark than light group. It might be interesting to discuss what kind of role the thalamus has in the visual pathway. For example, does it mainly have inhibitory connections, where less activity with light-stimulation leads to disinhibition of downstream areas?

We observed a spontaneous strong lateralization in the right thalamus of embryos incubated in the darkness, which differed significantly from embryos incubated with light, resulting in a ‘less’ right lateralized thalamus for light exposed group. Light stimulation at this embryonic stage is reaching only the right eye, visual input decussates at the level of the optic chiasm reaching the left thalamus. Therefore, we hypothesize that light stimulation is increasing brain activity in the left thalamus, decreasing the spontaneous level of asymmetry in this region. We have clarified this point in the revised manuscript (ll. 124-126).

– I think that the interpretation that more bilateral thalamic activation leads to more right-dominant wulst activity is not clear. I would assume that the contralateral projections would be smaller than the ipsilateral connections. If this is true, then more left-sided thalamic activity in the light (than dark) group should drive more activity in the left-sided wulst (while at the same time also the right-sided wulst, but not to the extent of net right-dominance).

Light exposure of chick embryo at this stage is known to cause an asymmetrical growth of fibers in the supraoptic decussation (L.J. Rogers, C. Deng, Light experience and lateralization of the two visual pathways in the chick. Behavioural brain research, 98(2), 277-287 (1999)). Such asymmetry consists in a higher density of fibers that from the left thalamus (more precisely Gld) reach the right visual Wulst (see Figure 1a). Hence, we hypothesize that the trend for right-lateralization in the visual Wulst reflects the asymmetry in the growth of fibers in the supraoptic decussation caused by asymmetrical light stimulation.

– Neural activity in the nucleus rotundus and/or entopallium should be included in the analyses to be able to discuss lateralization of the tectofugal pathway.

Reliable ROI segmentation are not easy to obtain, therefore for the present work, we limited our analysis to a restricted number of regions selected on the basis of our a priori hypothesis based on our species of interest, the domestic chick. Nevertheless, we are currently working on higher resolution acquisitions so that future studies might include other regions of interest. In the manuscript, we have clarified that our conclusions pertain to the tectum, and not necessarily to the entire tectofugal pathway. Meanwhile, we would like to disseminate the current protocol and the results it led to at an early stage to enable and encourage other researchers in the field.

– It is not clear from the introduction or figure 1A what pathways are measured and compared. Figure 1A should show the two pathways instead of only the thalamofugal one, or another figure could be added for the tectofugal pathway.

Based on previous evidence on lateralization of the visual system in the domestic chick, we focused on the thalamofugal visual pathway in this work, and added the optic tectum only as a control region.

– With the post-hoc t-testing, the authors should correct for multiple testing.

As detailed in our response to the public review,

we considered the reviewer's observation regarding the need for improvements in the statistical methods. In response, we have made amendments to the relevant section of the manuscript, explicitly stating that significant findings were obtained using a two-way ANOVA. For comparisons between conditions within specific brain regions, we conducted two-sample t-tests, and the results were corrected for Type I errors using the false discovery rate (FDR) method. One-sample t-tests were employed to assess lateralization across brain regions and conditions, and the corresponding p-values were reported without correction for multiple comparisons.

– Results should be compared to relevant literature, such as the following:Rogers, 2006, Cortex, doi: 10.1016/S0010-9452(08)70332-0Valencia-Alfonso et al., 2009, Phil. Trans. R. Soc. B, doi:10.1098/rstb.2008.0240Concha et al., 2012, Nat Rev Neurosci, doi:10.1038/nrn3371Manns and Ströckens, 2014, Frontiers Psych, doi: 10.3389/fpsyg.2014.00206

We thank the Reviewer for the kind suggestion, however Communications are limited to the number of 10 references, therefore we had to select the most essential ones. We have emphasized references restricted to the domestic chick, the species of investigation in the present manuscript.

– It is my understanding that projections from optical tectum to thalamus are symmetric if eggs are incubated in the dark (Gunturkun 1993; Rogers, 2006). This result from the literature should be cited, and it should be discussed how the current results relate to it.

We believe that the reviewer correctly refers to pigeon embryos (Güntürkün, O. (1993). The ontogeny of visual lateralization in pigeons. German Journal of Psychology.). In chicks instead, tectofugal pathway is not affected from light exposure during the sensitive window pre-hatching (Rogers, L. J., and Deng, C. (1999). Light experience and lateralization of the two visual pathways in the chick. Behavioural brain research, 98(2), 277-287.).